# Characterization and Optimization of Salep Mucilage Bionanocomposite Films Containing *Allium jesdianum* Boiss. Nanoliposomes for Antibacterial Food Packaging Utilization

**DOI:** 10.3390/molecules27207032

**Published:** 2022-10-18

**Authors:** Mohammad Ekrami, Ali Ekrami, Mohammad Ali Hosseini, Zahra Emam-Djomeh

**Affiliations:** 1Transfer Phenomena Laboratory (TPL), Department of Food Science, Technology and Engineering, College of Agriculture and Natural Resources, University of Tehran, Karaj 7787131587, Iran; 2Department of Nursing, University of Social Welfare and Rehabilitation Sciences, Tehran 1985713871, Iran; 3Functional Food Research Core, College of Agriculture and Natural Resources, University of Tehran, Karaj 7787131587, Iran

**Keywords:** *Allium jesdianum* Boiss., salep, nanoliposome, in vitro release, antibacterial activity, food packaging

## Abstract

**Highlights:**

**What are the main findings?**
Salep mucilage and NL_P_/A_EO_ were used to develop an antibacterial bionanocomposite film.A thin-layer hydration method was used to produce antibacterial NL_P_/A_EO_.

**What is the implication of the main finding?**
The incorporation of A_EO_ and NL_P_/A_EO_ affected surface morphology, physicochemical, barrier, and mechanical properties of the Salep mucilage bionanocomposite film.Encapsulating A_EO_ into NL_P_ decreased the A_EO_ release rate from the Salep mucilage bionanocomposite film.

**Abstract:**

This research aimed to characterize and compare the properties of nanoliposome (NL_P_)-loaded Salep mucilage-based bionanocomposite films containing free and encapsulated *Allium jesdianum* Boiss. essential oil (A_EO_). The mean size of nanoliposome containing *Allium jesdianum* Boiss. essential oil (NL_P_/A_EO_) was around 125 nm, the zeta potential value was about −35 mV, and the entrapment effectiveness was over 70% based on an evaluation of NL_P_ prepared using the thin-film hydration and ultrasonic approach. Morphological studies further corroborated the findings of the Zetasizer investigation. When NL_P_/A_EO_ has added to Salep mucilage-based bionanocomposite films, the tensile strength (TS), water solubility (WS), water content (WC), and water vapor permeability (WVP) were found to decrease. In contrast, the contact angle and oxygen permeability (O_2_P) elongation at break (EAB) increased. Scanning electron microscopy (SEM) and atomic force microscopy (AFM) images indicated that Salep mucilage-based bionanocomposite films added with NL_P_/A_EO_ had a disordered inner network in the cross-section and a rough structure on the surface compared to the control film. Finally, an increase in antibacterial activity and a decrease in A_EO_ release rate was observed for the Salep mucilage-based bionanocomposite films incorporated with NL_P_/A_EO_. Our results indicated that NL_P_/A_EO_, as an innovative sustained-release system, had the potential for using the developed antibacterial food packaging base on Salep mucilage for the shelf life extension of perishable food products.

## 1. Introduction

Biopolymer films have seen a significant uptick in their usage over the last two decades as a means of extending the shelf life of products due to the environmental benefits they provide over traditional synthetic polymeric films. The term “edible film” refers to any thin coating that may be placed that serves as a barrier to scent, moisture, and oxygen, whether it be based on hydrocolloids, lipids, or a mixture of the two (composites) [1]. Mucilages, as extractive polysaccharides of plant sources, are of particular interest to scientists because of the abundance of their resources, the quality of their film preparation, and the low permeability of their film to gases [2]. Salep was one of our previous research’s inexpensive, accessible, and easy-to-use sources for developing bionanocomposite edible films.

Salep, flour milled from ground orchid tubers, contains humidity (12%), protein (5%), starch (2.7%), ash (2.4%), and is a rich source of glucomannan (16–55%) [3]. Salep glucomannan consists of glucose and mannose in the ratio of 1:3.8, with a linear (or slightly branched) chain in which the hexopyranose groups are bound via β-(1→4) linkages. Because of its polymeric structure, aphrodisiac effect, and other medicinal characteristics, Salep is known as a food and pharmaceutical material [4].

Various food preservatives may be used as antimicrobial agents to keep food fresh for extended periods [5]. It has been more popular to protect food using natural preservative items, such as herbal extracts and essential oils (EOs), rather than chemical food preservatives. Plant EOs have shown some practical inhibitory effects on foodborne diseases [6].

Among the plants of the genus Allium (family Amaryllidaceae), *Allium jesdianum* Boiss. (Bon-e-Sorkh in Persian) is one of the main species of this genus. It is an endemic Iranian plant naturally grown in Iran’s northern, western, and southwestern regions [7]. Iranians have used its medicinal and antimicrobial features from ancient times. This plant is used in diet and, of course, in traditional treatment for digestive pains, like rheumatic pains [8]. This remarkable genus has been linked to various potential health benefits, including antibacterial and anticancer effects [6]. Thus, several initiatives have isolated its functional therapeutic components [9]. However, the effects of A_EO_ on nanostructures as herbal drug delivery vehicles are little understood.

Nanostructure-based targeting strategies have been suggested to overcome microbial resistance as EOs have a rapid decomposition rate, biological fluctuation, and minimal biological presence. Biochemical, pharmaceutical, and agricultural objectives may be delivered using lipid-based carrier systems such as the nanoliposome (NL_P_) [10]. By employing NL_P_, it is possible to focus on a particular target. There must be sufficient concentrations of certain bioactive substances at the target location for optimal effectiveness, either in vitro or in vivo. Improved cost-effectiveness might be achieved by prompt and targeted release of bioactive compounds, a wider variety of applications, and optimal dosing [11]. The Lipid vesicles aim to use both active and passive techniques possible As they are made up of a fat outer layer, they can protect, encapsulate, and transport lipophilic medications throughout the body. In addition to being biodegradable and immune-free, they may serve as reservoirs for the slow-release medications which are enclosed inside them [12]. Thus, they are great EO carriers. Lipids and water may be kept together in the same environment and protected from damage and degradation by using NL_P_/based delivery methods, which lead them to be both lipophilic and hydrophilicsimultaneously [13]. A slight safety is expected by combining it with natural materials. [10]. In medicine, emphasis is being paid to these products because of their low toxicity, drug distribution modifications, pharmacokinetics, and enhanced therapeutic index [14]. Research in this area has focused on designing and testing innovative NL_P_/based medication delivery strategies. Increased solubility, bioavailability, stability, pharmacological activity, and decreased toxicity and adverse effects are advantages of employing plant compounds in nano-liposome delivery methods [11,15].

As a food-grade delivery mechanism, liposomes have yet to be thoroughly studied. Notwithstanding the excellent antimicrobial and antioxidant nature of essential oils (EO)/or extracts, their direct usage is often limited due to strong odor, inability to dissolve in liquid foods, susceptibility to heat, alkaline conditions, and destruction by environmental influences during food preparation and storage [16]. Consequently, incorporating them into films’ structure has been an attractive approach to overcome these problems. Hence, nanotechnology and nanoencapsulation of essential oils/extracts by using suitable polymeric microand nanoparticle systems could be one of the potent approaches suggested to overcome these deficiencies [17]. Recently, films and biocomposites have been extensively studied for their potential ability to encapsulate nanoliposome structures. As a result, films are considered an ideal carrier for encapsulating bioactive components. In this regard, Homayounpour et al. (2021) developed nanochitosan-based active packaging films containing free and nanoliposome caraway (*Carum carvi*. L.) seed extract [17]. Chavoshi et al. (2022) prepared and characterized *Psyllium* seed gum films loading *Oliveria decumbens* essential oil encapsulated in nanoliposomes [18]. In other research, Ghasempour et al. (2022) optimized and utilized Persian gum/whey protein bionanocomposite films containing betanin nanoliposomes for food packaging utilization [19].

This study aimed to create NL_P_ incorporating A_EO_ for developed Salep mucilage-based bionanocomposite films as a sustained release system for antibacterial food packaging utilization.

## 2. Experimental

### 2.1. Materials

*Allium jesdianum* Boiss. was collected from the highlands regions of Shahr-e Kord, Iran. The palmate tuber Salep (Figure 1) was acquired from Sanandaj, Iran. L-α-lecithin of soybean (94% phosphatidylcholine), cholesterol (grade ≥99%), 2,2-di-phenyl-1-picrylhydrazyl (DPPH) obtained from Sigma-Aldrich, St. Louis, MO, USA. Glycerol was obtained from ACROS, Loughborough, UK) and used as a softener. All reagents, organic solvents, and chemicals (including dichloromethane, methanol, ethanol, and phosphate buffer saline (PBS), supplied from Merck, Darmstadt, Germany) were of analytical grade. Quelab, Canada supplied Brain Heart Infusion (BHI) broth used for microbiological evaluations). The whole of the chemicals was used in primary form, with no effect of further purification. The lyophilized bacterial strains (*Escherichia coli O157:H7* (PTCC No. 1330) and *Staphylococcus aureus* (PTCC No. 1431)) were obtained from a Persian-type culture collection center (PTCC) in Tehran, Iran. Chloramphenicol as a positive control in antibacterial activity assay purchased from BioBasic, Canada).

### 2.2. Essential Oil Extraction and Characterization

According to our earlier research [20], the freshly washed and finely dried samples were hydrodistilled in a glass Clevenger-type equipment for 3 h. An anhydrous sodium sulfate-dried oil with a pleasant aroma was kept in sealed vials at 4 °C for further GC/MS analysis. Hewlett-Packard GC/MS (Perkin Elmer Auto System XL), also equipped with an MS, was employed for the analysis. A DB-1 capillary column (30 m 0.25 mm ID 0.25 mm film thickness) was used in the chromatograph. Helium was used as a carrier gas, with a 1.5 mL/min flow rate, to inject 1 L of 0.1% A_EO_ solution. As a starting point, the column was heated to 50 °C for 6 min before being raised to 240 °C at a rate of 3 °C/min, then sustained at 300 °C for 3 min. In the electron ionization mode of major spectrometry, the ionization voltage was 70 eV, and the source temperature was 270 °C. We used the literature’s retention indices and mass spectra to compare our samples and identify the ingredients. 

### 2.3. Nanoliposome Preparation and Characterization

Our earlier procedure [13] showed that NL_P_/A_EO_ was prepared to utilize thin-film hydration. A combination of lipids containing lecithin: cholesterol: A_EO_ (3:1:1) was diluted with dichloromethane/methanol (1:1) and then added to a flask with a circular bottom (250 mL). The solvent evaporated and caused the constantly spinning rotary evaporator, leaving a thin film on the vacuum oven’s 24-h-dried, 60 °C inside walls. The lipid film is hydrated in 50 mL of PBS at pH = 7.4, and the mixture is stirred for 1.5 h at a temperature above the gel-liquid transfer temperature. Next, the mixture is sonicated at (4 °C, 30 s—1:1 on/off, 24 kHz, 90% power) using a probe-type sonicator (Hielscher, Germany) and then filtered through a 0.22 mm membrane. 

#### 2.3.1. Determination of Size and Zeta Potential

The mean particle size (z-average in intensity), polydispersity index as size distribution, and zeta-potential of liposome dispersions were measured using a Zetasizer (Malvern Instruments Ltd., Worcestershire, UK). For this purpose, a sample was injected into a capillary tube, diluted with distilled water 10 times, and placed in a designated chamber at pH 7.4 and 25 °C. 

#### 2.3.2. Encapsulation Efficiency 

Degradation and separation by centrifuge were used to assess efficiency [21]. Using a centrifuge (Hettich, Germany) at 2500× *g* for 10 min, nanoliposomes containing EOs were separated from unencapsulated materials (Hettich, Germany). The liposomes were destroyed in ethanol 90% (Merck, Darmstadt, Germany), and encapsulated EO content was measured using a UV/Visible spectrophotometer (Cecil, MA, USA). Since the spectrophotometric spectrum for free A_EO_ provides a peak with maximum intensity at λ = 275 nm, the measurements were performed at this wavelength. The efficiency of encapsulation was determined by the following Equation (1):(1)$$\mathrm{Encapsulation}\;\mathrm{efficiency}=(D_{L}/D_{T})\times 100$$
where *D_L_* is drug-loaded, and *D_T_* is a total drug used.

#### 2.3.3. Morphology

Scanning electron microscopy (SEM) (Philips XL30, Eindhoven, The Netherlands) was used to assess NL_P_/A_EO_’s microstructure. Before visualization, the NLP/AEO was fixed on an aluminum stub using adhesive tape and spluttered with gold using a sputter coater (Balzers, Liechtenstein). The observation was conducted at 15 kV acceleration potential. 

NL_P_/A_EO_ were examined by tapping mode atomic force microscopy (AFM) (Nanoscope IIIa Multimode, Santa Barbara, CA, USA) equipped with an E-type scanner. The rectangular silicon cantilever was employed with a nominal spring constant of 5–100 N/m and nominal resonance frequencies of 10–320 kHz. The NL_P_/A_EO_ were attached to mica surfaces and examined in the air at 25 °C with 65% relative humidity. NanoScope software (NanoScope Analysis v140r1sr2) is employed for all AFM image processing. 

### 2.4. Preparation of Bionanocomposite Films and Characterization

The Salep mucilage edible control films were prepared using our previous method reported [22]. Initially, yellow-white Salep powder was obtained from palmate tuber Salep after washing for 15 min with distilled water, drying at 40 °C in the oven for 24 h at atmospheric pressure, and milling. Then, the Salep powder was immersed in distilled water with a mixer (IKA, Staufen, Germany) at 25 °C for 1 h. The mixture was homogenized at 10,000× *g* for 5 min using a homogenizer (IKA, Staufen, Germany); a centrifuge at 3000× *g* for 10 min solutions was filtered. The mucilage was dried by a freeze-dryer (Martin Christ, Osterode am Harz, Germany) and stored in the dark at 4 °C for further use. Film solution was prepared by slowly dissolving 2% (*w/v*) mucilage in distilled water and adding glycerol (25% *w/w* based on Salep mucilage weight) as a plasticizer under constant stirring (500 rpm) at 25 ± 2 °C for 1 h. At the same time, A_EO_ and NL_P_/A_EO_ (2.5 wt.% based on solution dry matter) were added and stirred at 25 °C for 30 min to obtain a homogeneous solution. After that, the film-forming solution was poured onto plastic Petri dishes, dried at 35 °C for 24 h, and then conditioned at 25 °C—53% RH (by placing a saturated magnesium nitrate solution), the control and antibacterial bionanocomposite films (Figure 1) were ready for use. 

#### 2.4.1. Color Parameters 

Color parameters of the bionanocomposite films were taken digitally, and their colors were evaluated using an instrumental colorimeter (Minolta, Japan) after being calibrated against a white plate. Using the color coordinates L* or lightness (black = 0 to white = 100), a* (greenness = −60 to redness = +60) and b* (blueness = −60 to yellowness = +60), the following equations are used to calculate the total color difference (ΔE) (Equation (2)), whitish (WI) (Equation (3)), and yellowness (YI) (Equation (4)) indices of samples, respectively [20]: (2)$$\Delta E=\sqrt{{\left(L-L^{\prime }\right)}^{2}+{\left(a-a^{\prime }\right)}^{2}+{\left(b-b^{\prime }\right)}^{2}}$$
(3)$$\mathrm{WI}=100-\sqrt{{\left(100-L\right)}^{2}+a^{2}+b^{2}}$$
(4)YI=142.86×b/L

#### 2.4.2. Physical Properties

##### Film Thickness

The thickness of the bionanocomposite films made from Salep mucilage was measured using a digital micrometer (Mitutoyo, Japan) by selecting ten random sites with a resolution of 0.001 mm. 

##### Moisture Content (MC)

Salep mucilage bionanocomposite films were conditioned at room temperature and 53% RH for 24 h before their MC was measured. Before being dried at 110 °C for 24 h, the bionanocomposite films were weighed and placed in a pre-weighed aluminum capsule. The following Equation (5) was then used to calculate MC: (5)$$\mathrm{MC}\;\left(\%\right)=\left(W_{i}-W_{d}/W_{i}\right)\times 100$$
here, *W_i_* is the initial sample weight, and *W_d_* is the dried sample weight.

##### Water Solubility (WS)

To calculate the WS of the Salep mucilage bionanocomposite films, it all started with 24 h drying time at 110 °C for 3 × 3 cm^2^ films. After the bionanocomposite films were weighed to establish their dry weight, they were submerged in 40 mL of water and agitated for 24 h at room temperature. Separated and dried at 110 °C, the ultimate weights of insoluble bionanocomposite films were determined. The following Equation (6) was then used to determine their WS: (6)$$\mathrm{WS}\;\left(\%\right)=\left(W_{i}-W_{f}/W_{i}\right)\times 100$$
here, *W_i_* is the initial sample weight, and *W_f_* is the insoluble sample weight [23].

##### Transparency

The transparency value of the Salep mucilage bionanocomposite films was measured by using a UV/Visible Spectrophotometer according to ASTM D1746 standard method [24] and calculated from the following Equation (7):(7)$$T_{600}=\left(\log T\%/L\right)\times 100\;$$

*T*_600_ is the transparency value at 600 nm, *T*% is the transmittance percentage at 600 nm, and *L* is the film thickness (mm). All measurements of color properties were conducted in three replications.

##### Wettability

Contact angle (CA) was measured using the sessile drop technique to establish the wettability of Salep mucilage bionanocomposite films. An optical goniometer (Kruss, Germany) was used to determine the contact angle between the surfaces of the produced films and water. To accomplish this, we placed a droplet of distilled water (4 μL) on the films and measured the droplet’s angle. 

#### 2.4.3. Permeability Properties

##### Water Vapor Permeability (WVP) 

The Salep mucilage bionanocomposite films’ WVP was measured using a tweaked version of ASTM E96-95s procedure [25]. Before testing, the films were conditioned in a laboratory for 24 h at ambient temperature and 53% RH. As a result, we may derive the WVP (g m^−1^ s^−1^ pa^−1^) using Equation (8):(8)WVP=(Δw×X)/(t×A×Δp)
here, Δ*w* is the weight change of the glass cell containing the water (g), *X* is the average bionanocomposite films thickness (m), *t* is the time (s), *A* is the area of the exposed bionanocomposite films (m^2^), and Δ*p* is the water vapor pressure difference across the two sides of the bionanocomposite films (The difference in RH between assay cup (calcium-chloride desiccant 0% RH) and saturated solution of sodium chloride (75% RH) corresponded to a driving force of 1753.55 Pa, expressed as water-vapor partial pressure) [20].

##### Oxygen Permeability (O_2_P)

The O_2_P of the bionanocomposite films was measured at 25 °C and 53% RH using a gas permeability tester (Brugger, Munich, Germany) according to the standard ASTM D3985. Accordingly, the oxygen permeability is calculated as follows Equation (9):(9)$$O_{2}P=\left(\mathrm{OTR}\times X\right)/\Delta p$$
where O_2_P is oxygen permeability (cm^3^ µm m^−2^ day^−1^ kPa^−1^), OTR is oxygen transmission rate (cm^3^ m^−2^ day^−1^), *X* is the film thickness (µm), Δ*p* is the oxygen partial pressure difference across the film (101 kPa) [26]. 

#### 2.4.4. Mechanical Properties 

A texture analyzer (Testometric, Rochdale, UK) was used to determine the tensile strength (TS) and elongation at break (EAB) of Salep mucilage bionanocomposite films according to the standard method of ASTM D882 [27]. Before testing, the films were previously conditioned (53% RH at 25 °C for 48 h). Rectangular (10 × 100 mm) strips of samples were tested at a crosshead speed of 0.5 mm/s. 

#### 2.4.5. Release Properties

The release of A_EO_ of NL_P_/A_EO_ and A_EO_-loaded Salep mucilage bionanocomposite films at different temperatures (4, 25, and 37 °C) was evaluated utilizing the cumulative release method with a slight modification described by Aziz and Almasi (2018) [28]. Therefore, a dialysis tube in a 100 rpm shaking bath was employed to assess the release of A_EO_ in vitro in 100 mL ethanol (95 % *v/v*). Additionally, the released NL_P_/A_EO_ was measured regularly by a UV/Visible spectrophotometer (λ = 275 nm). The cumulative percentage of A_EO_ released was determined by the following Equation (10):(10)$$\mathrm{Cumulative}\;\mathrm{release}\;\left(\%\right)=\overset{t}{\underset{t=0}{\sum }}\frac{M_{t}}{M_{0}}\;\;\;\;\;$$
where M*t* is the cumulative amount of A_EO_ released at each point in the sampling time, and M_0_ is the initial weight of A_EO_ loaded in the sample.

#### 2.4.6. Antibacterial Properties

*Escherichia coli O157:H7* (PTCC No. 1330) and *Staphylococcus aureus* (PTCC No. 1431) were used as model food spoilage or pathogenic microorganisms. The bacteria were incubated overnight at 37 °C on Brain Heart Infusion (BHI) broth. The antibacterial activity of the Salep mucilage edible indicator films was then determined using the agar disc diffusion assay, according to the method described by Ekrami et al. (2019) [20], with some modifications. Suspensions of the microorganisms (1–5 10^8^ CFU/mL) were placed on the surfaces of the BHI agar plates. Measurements of the dimensions of the inhibition zones were used to establish the antibacterial efficacy of the different microbial strains. For this experiment, 6 mm diameter disks of films were cut and sterilized by UV irradiation for 30 s, then placed on MHA inoculated agar, then incubated at 37 °C for 18 h. Chloramphenicol (a broad-spectrum antibiotic including several Gram-positive and Gram-negative bacteria) was selected as a positive control and NL_P_ as a negative control in this assay.

## 3. Results

### 3.1. Identification of A_EO_ Components

A_EO_s chemical composition accounted for 0.19% of light brown EO production. The discovered compounds made up a total of 94.62% of AEO (Table 1). The main constituents of A_EO_ are organic trisulfide, with Dimethyl trisulfide (17.35%), Dimethyl tetrasulfide, and (11.84%) Dipropyl trisulfide (8.11%) as the main components. Other vital constituents which are present in the volatile oil are Hexadecanoic acid (7.77%) as saturated fatty acid, Neral (Z-Citral) (6.20%) as an enal and a monoterpenoid, and Pentacosane (5.81%) as an alkane. The main component of A_EO_ identified in the present research is in excellent agreement with their findings [29]. However, the principal chemical components of the genus *Allium* are found in various sources significantly. Genetics, age and maturity, weather, soil composition, plant organs, distillation conditions, and other variables may affect these variations [30].

### 3.2. Characterization of NL_P_/A_EO_

#### 3.2.1. Determination of Size, Zeta Potential, and Encapsulation Efficiency

The results of these three tests on 10 mg/mL NL_P_/A_EO_ are shown in Figure 2. The PDI of 0.383 was found for particles 124.7 nm in size (Figure 2A). Zeta potential for NL_P_/A_EO_ was −35.8 ± 3.2 mV. (Figure 2B). In this research, tiny, highly zeta-potential liposomes were intentionally created. Smaller liposomes (200 nm) are more likely to effectively use the entrapped molecule, as proposed [31]. The surface zeta potential may significantly affect the dispersibility and stability of liposome systems. Liposomes are more stable when the zeta potential is more remarkable because the particles are repelled from one another [32]. These findings suggest that the manufactured liposomes are stable and dispersible and can entrap sufficient quantities of bioactive substances. Incorporating A_EO_ into the liposome membrane’s ordered structure of lecithin boosts the membrane’s fluidity and thickness. The average particle size is affected by factors such as viscosity, lipid/bioactive ratios, the kind and amount of stabilizing substances like cholesterol, manufacturing procedures, and preparation techniques, including film layer thickness and temperature [33]. 

The addition of bioactive material may increase particle size. If the bioactive ingredient could be more densely packed, the sphere would have a smaller diameter [34]. The encapsulation efficiency was 70.09 ± 6.18%. The A_EO_ loading saturation in NL_P_ and the limited amount of nanoliposome vesicles accessible for A_EO_ were other vital factors in encapsulation efficiency. According to prior research, as long as the quantity of drug in the capsule is less than the limit, the loading of the drug rises, but if the amount of medicine in the capsule exceeds the limit, the loading decreases [35].

#### 3.2.2. Morphology

NL_P_/A_EO_ evaluation was performed based on SEM and AFM findings. Figure 3A indicates that NL_P_/A_EO_ is spherical, multidimensional, and integrated vesicles with a very narrow distribution, as shown in the AFM micrograph. In the SEM micrograph, NL_P_/A_EO_ resulted in spherical and single dispersed particles (Figure 3B). Neither aggregation nor coalescence was seen. The morphological micrographs show that the NL_P_ has a particle size of about 112 nm, which agrees with the particle size determined by DLS analysis.

### 3.3. Performance Analysis of the Film

#### 3.3.1. Color Parameters

Table 2 displays the values for color parameters (L, a, b, ΔE, WI, YI, and transparency) for NL_P_/A_EO_ and A_EO_-loaded bionanocomposite films. Adding the NL_P_/A_EO_ and A_EO_ has a negative effect on transparency, L, and WI (*p* < 0.05) but a positive effect on ΔE, a, b, and YI. In most cases, the antibacterial films became darker and somewhat yellower after being treated with NL_P_/A_EO_ and A_EO_ (Figure 1). The addition of EO and NL_P_ caused emulsified films to become more opaque; this was likely owing to an increase in diffuse reflectance triggered by light scattering in the lipid droplets; the less intense the light scattering, the lower the WI of the film [36]. Similar behavior has been recorded by Jouki et al. (2014) for the incorporation of thyme EO into quince seed mucilage films [37].

#### 3.3.2. Physical Properties

According to Table 3, after adding NL_P_/A_EO_ and A_EO_, the thickness of the bionanocomposite film increased significantly (*p* < 0.05). The films prepared with NL_P_/A_EO_ and A_EO_ represent lower moisture content and solubility than the Salep control film. The result is mainly attributed to reducing the polymer matrix’s hydrophilic properties for the sake of the hydrophobic nature of A_EO,_ which leads to an increase in surface roughness and a specific size, and the distribution of NL_P_/A_EO_ in the film network led to an increase of the film’s volume.

Similar results were found by Ekrami et al. (2019) for Salep edible films functionalized with pennyroyal (*Mentha pulegium*) EO [20] and Wu et al. (2015) for fish gelatin films incorporated with cinnamon EO nanoliposomes [35]. 

The NL_P_/A_EO_ and A_EO_-loaded bionanocomposite films had lower moisture content and water solubility than the control films (Table 3). A_EO_ includes hydrophobic substances such as saturated fatty acid [29]; for this reason, an increase in water repellency due to the presence of EO can be expected. Researchers have reported that incorporating eucalyptus oil into chitosan films decreased water solubility [38]. The lower water holding capacity could be attributed to the interactions between NL_P_ and hydrophilic sites of the Salep chain, which substituted the interactions of the A_EO_-water solution that predominated in the control film. Other researchers reported that the moisture content of fish gelatin films decreased after incorporating cinnamon EO nanoliposomes, attributed to increased film hydrophobicity, reducing water absorption [35].

Film opacity was boosted by using NL_P_/A_EO_ and A_EO_ (Table 3), although the effect of free EO was more than that of encapsulated EO. Two reasons contribute to this result: the encapsulation conditions used to enclose the A_EO_ reduce the scattering and absorption of light. Moreover, the hydrophobic material in A_EO_ creates light-scattering colloidal particles in the film.

A lower contact angle shows the incorporation of NLP/AEO and AEO (Table 3). Surfaces of films are considered hydrophilic when their contact angles are less than 90° and hydrophobic when they are more than 90°. The comparatively high polarity of the functional groups on the Salep mucilage molecules explains the reason for the hydrophilic control films (~68°). When observing the wettability by measuring the contact angle of NL_P_/A_EO_ and A_EO_-incorporated Salep mucilage bionanocomposite films, we found that they are hydrophobic, with a contact angle higher than 90°. One theory is that the increased connection with the A_EO_ makes the hydrophilic groups on the sheet surface less accessible, hence decreasing the number of polar sites that may form a hydrogen bond with the water droplet [39]. The EOs non-polar molecules attached to the biopolymer molecules’ surfaces are responsible for the increased hydrophobicity of the resulting films [40].

#### 3.3.3. Mechanical Properties

To determine the exact effect of NL_P_/A_EO_ and A_EO_ addition on the mechanical characteristics of Salep mucilage bionanocomposite films, compared to the tensile strength and elongation at the break (Table 3). Even though the tensile strength of the control films dropped from 18.90 ± 1.23 MPa—82% after adding NL_P_/A_EO_ and A_EO_ up to 16.37 ± 1.02 MPa—103% and 11.46 ± 1.15—63%, respectively.

Changes in the films’ mechanical characteristics may be traced back to alterations in the cross-linking of the biopolymers brought about by the chemicals in the A_EO_. For example, the many hydroxyl groups in anthocyanins may create hydrogen bonds with gelatin, reducing the number of cross-links between the gelatin molecules. This fact implies that the films may be stretched farther without tearing, yet with less effort. Other studies have documented a comparable drop in tensile strength and an increase in elongation at break when adding EOs, stearic acid, and palmitic acid to gelatin films [41]. It is worth noting that the films also contained glycerol, a food-grade plasticizer that enhances the mobility of polymer chains and so reduces tensile strength; therefore, anthocyanin can replace it to some extent.

#### 3.3.4. Permeability Properties

The O_2_P and WVP of the Salep mucilage bionanocomposite films were also checked (Figure 4). The WVP of the antibacterial films incorporated with NL_P_/A_EO_ and A_EO_ was slightly decreased to 4.29 and 18.98 × 10^−12^ g s^−1^ m^−1^ Pa, respectively. The WVP is affected by factors such as the hydrophilic-hydrophobic ratio of film components and the hygroscopic nature of the anthocyanins utilized [42]. So, adding hydrophobic chemicals like EO and lipids improves the film’s hydrophobicity and structural complexity, leading to more excellent water resistance [43]. In confirmation of the effect of EO on WVP, Ghasemlou et al. (2013) reported that the lower WVP of starch films containing *Zataria multiflora Boiss* or *Mentha pulegium* EO may be due to hydrogen and covalent interactions between the starch lattice and these polyphenolic compounds. These interactions can limit the availability of hydrogen groups to form hydrophilic bonds with water, leading to a decrease in the film’s affinity for water [44].

The O_2_P of the salep control film was 30.61 ± 1.60 cm^3^ µmm^−2^ d^−1^ kPa^−1^ (Figure 4). The films and hydrophilic coatings (polysaccharides or proteins) usually have an excellent barrier effect on oxygen transference [45]. Probably because of the even distribution of the hydrophobic section in the film’s structure, this attribute is considerably impacted by the addition of NL_P_/A_EO_ and A_EO_ to the Salep mucilage films. Larger pores can explain this outcome by increasing the non-polar phase in the polymer network [46]. This increase in film porosity may have been due to a weakening of the attractive interactions between the biopolymer molecules in the presence of the EO and lipids. Jouki et al. (2013) reported that quince seed mucilage films containing thyme EO exhibit relatively poor oxygen barrier properties. The more excellent solubility of oxygen can explain this in the non-polar oily phase, which increases the transfer rate of the oxygen molecules in the plasticized polymer matrix [37].

#### 3.3.5. In Vitro Release 

Figure 5 displays the A_EO_ release profiles from the Salep mucilage bionanocomposite films that were either A_EO_-loaded or NL_P_/A_EO_-loaded. When comparing the two types of films, the fractional mass release, calculated by dividing the amount of A_EO_ released at times by the total quantity of antimicrobial released at infinite time, was progressive and tended asymptotically to 1. Free EO showed quicker A_EO_ release than nanoliposomal EO at all tested temperatures (4, 25, and 37 °C). Including NL_P_/A_EO_ into Salep mucilage bionanocomposite films may enhance the time-controlled release and increase food shelf life. With the addition of NL_P_/A_EO_ to the film-forming solution, the hydrophilic groups of the Salep solution combine with the nanoliposomes’ surface to create a protective sheath, therefore increasing the nanoliposomes’ stability and hence extending the time it takes for the medication to be released. As a result, the pace at which NL_P_/A_EO_ was released into the model liquid was slowed. Even at 4 °C, the effect could be shown (with the same time-of-release quantity being lower at 4 °C). The liposome structure uses lecithin, which tends to gel at low temperatures, making A_EO_ release challenging. The electrostatic agent had a smaller dispersion coefficient at higher temperatures between the film and the liposome membrane. The lipid agent was more soluble in ethanol at higher temperatures. As a result, there was a more significant input of A_EO_ into the stimulating medium. Reduced A_EO_ release in the model fluid suggests that this NL_P_s antibacterial activity is responsible for its beneficial effect on drug release delay [47]. 

Increased A_EO_ affinity for NL_P_ may be achieved with a higher emulsifier: EOs ratio, while increased porosity of the NL_P_ matrix can be achieved with a higher A_EO_ ratio, and decreased mean size can be achieved with a lower diffusion rate inside the NL_P_ matrix [47,48,49,50,51]. 

The antimicrobial stability result agreed with the in vitro release result, suggesting that the A_EO_ incorporated in the vesicle of NL_P_ gave the film a prolonged release action and enhanced its antimicrobial stability. Similar results have been reported by Aziz and Almasi (2018), who studied the release properties of whey protein isolate films incorporated with thyme (*Thymus vulgaris* L.) extract-loaded nanoliposomes [28].

#### 3.3.6. Antibacterial Activity

Food poisoning pathogens were examined using the agar disk diffusion technique to determine the antibacterial activity of A_EO_ and NL_P_/A_EO_. Because the inhibition zones of NL_P_/A_EO_ are greater than those of A_EO_, the findings shown in Table 4 are more significant and effective when applied to the tested strains of microorganisms. Both compounds spread on agar plates and block the development of bacteria, particularly *Staphylococcus aureus* strains, which give the largest inhibition zones for A_EO_ and NL_P_/A_EO_. Microdilution was utilized to corroborate prior findings, and the MIC and MBC for A_EO_ and NL_P_/A_EO_ were found. Gram-positive bacteria have a different cell wall structure than gram-negative bacteria. Thus EOs have a little greater impact on them [52,53,54].

EOs may be protected against oxidation, light reactions, and high temperatures via encapsulating techniques [55]. The antimicrobial agent is transported across the cell membrane and released into the cytoplasmic membrane, where it may function. Antimicrobial chemicals are stored in the delivery system as a reservoir, ensuring a stable concentration of these compounds in the aqueous bulk phase throughout time [56]. Since EOs are poorly soluble in water, loading them into a limited aqueous environment would reduce the overall quantity of antimicrobial compounds available for a long time before they were consumed or degraded [57]. When loading antimicrobial compounds into emulsion droplets distributed in water, the concentration of bioactive compounds might be substantially more significant than the water solubility concentration, enabling a larger quantity to be loaded into the limited aqueous solution. When using an emulsion, the bioactive molecules’ concentration in the water phase would be maintained at a steady level for an extended period due to the EO component’s aqueous partitioning and the inclusion of an emulsifier [58,59]. The nanoencapsulation technique enhances the biological activity of EOs, such as antibacterial characteristics, by improving the surface-to-volume ratio by lowering the particle size to the nanoscale and boosting the bioavailability of EOs [60].

## 4. Conclusions

*Allium jesdianum* Boiss. essential oil was successfully extracted, characterized, and encapsulated in nanoliposomes using the thin-film hydration and ultrasonic approach. These liposomes showed high stability, an average size of ~125 nm, and relatively high encapsulation efficiency (~70%). The results of the present study indicate that NL_P_/A_EO_ can be incorporated into Salep mucilage-based bionanocomposite films without apparent loss of vesicle structural integrity but causing internal discontinuities in the film matrix. Adding the A_EO_ in the form of NL_P_ promoted transparency, tensile strength, O_2_P, WVP, in vitro release, and antibacterial activity against food poisoning pathogens (after 30 days) compared to the free A_EO_. While A_EO_ bioactive characteristics were being preserved throughout storage, the nanoliposome system proved an effective antibacterial. Additionally, this system is a natural antibacterial and preservative in the food and pharmaceutical sectors.

## Figures and Tables

**Figure 1 molecules-27-07032-f001:**
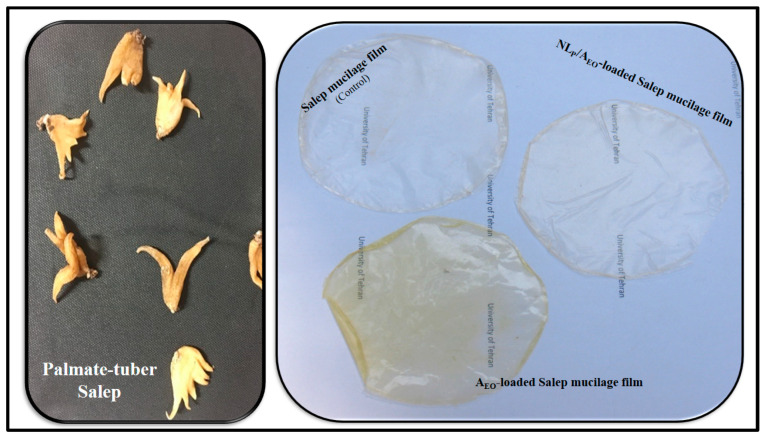
Photographic images of the palmate-tubers Salep and Salep mucilage control/antibacterial bionanocomposite films.

**Figure 2 molecules-27-07032-f002:**
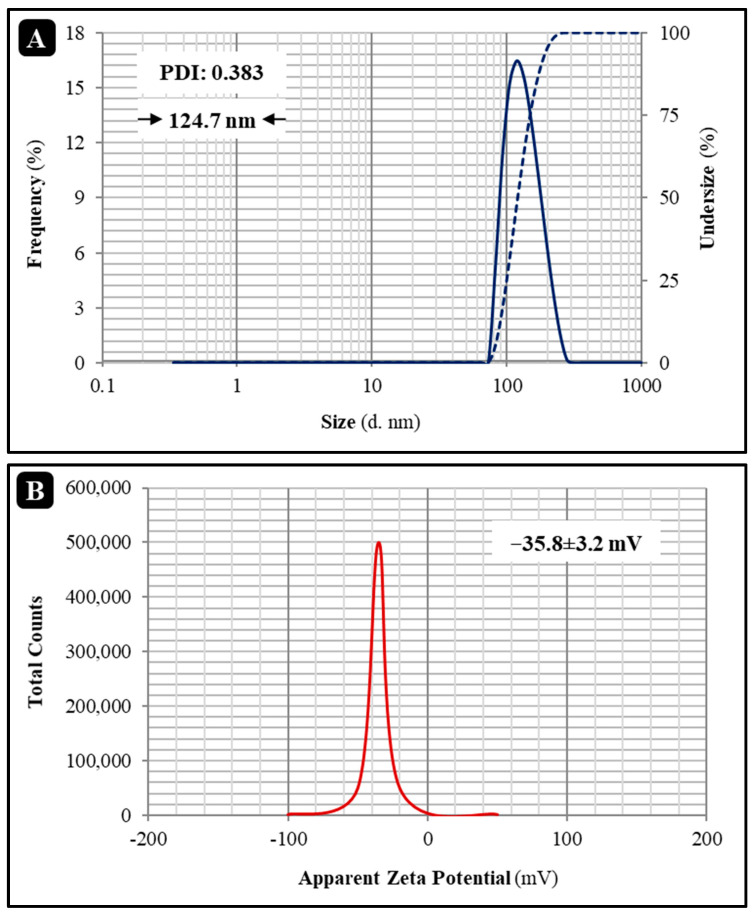
The size distribution (**A**) and zeta potential distribution (**B**) of *Allium jesdianum* Boiss. EO nanoliposomes.

**Figure 3 molecules-27-07032-f003:**
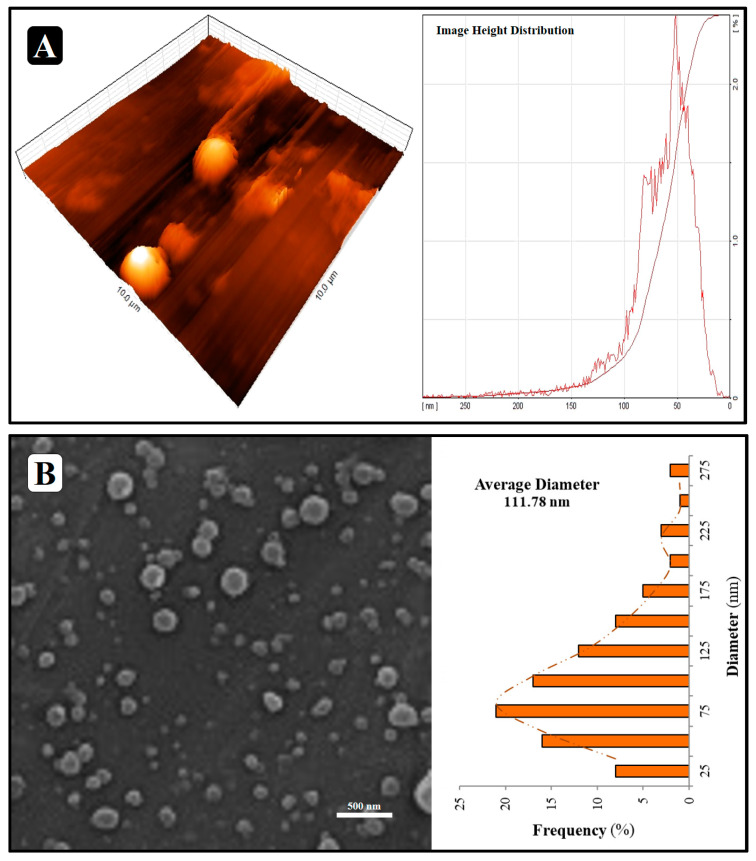
AFM (**A**) and SEM (**B**) micrographs of NL_P_/A_EO_.

**Figure 4 molecules-27-07032-f004:**
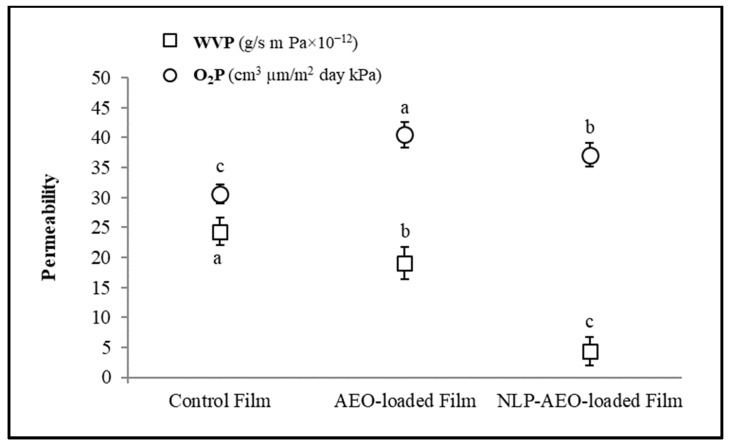
Effect of free and nanoliposomal A_EO_ on permeability properties of Salep mucilage bionanocomposite films.

**Figure 5 molecules-27-07032-f005:**
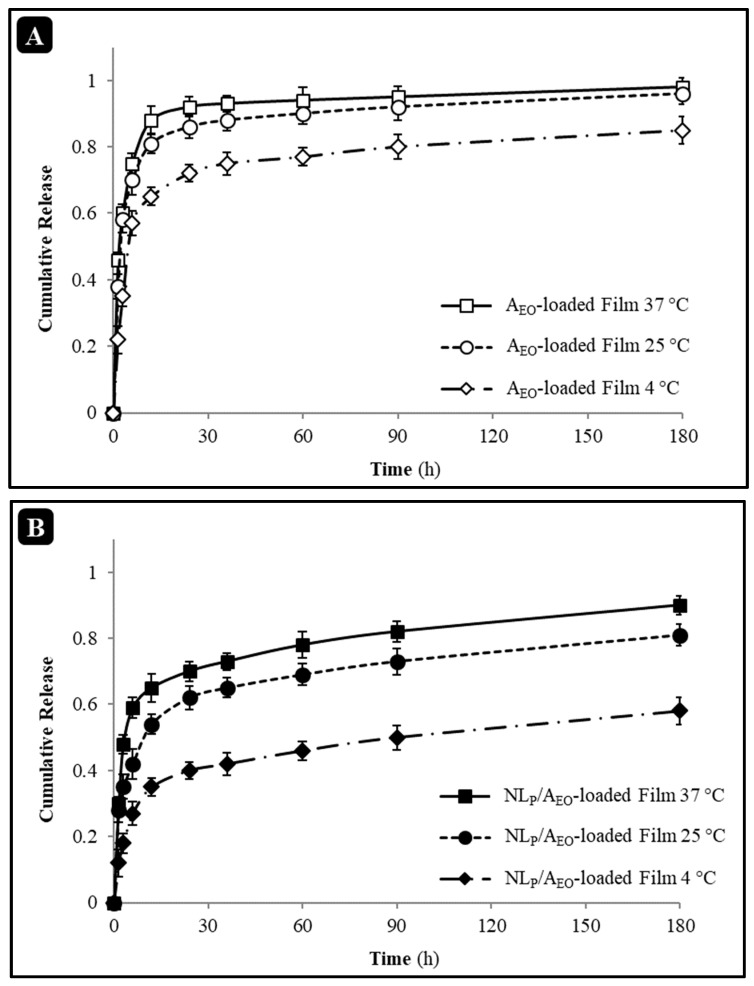
Effects of different temperatures on in vitro release profile of free A_EO_ (**A**,**B**) NL_P_/A_EO_-loaded Salep mucilage films.

**Table 1 molecules-27-07032-t001:** Chemical composition of *Allium jesdianum* Boiss. *EO*.

Compound	RI *	Concentration (%) **
Hexenal	777	0.24
1-Heptanal	904	0.12
Dimethyl trisulfide	963	19.05
Dehydroxy-trans-Linalool oxide	996	0.29
Benzene acetaldehyde	1039	0.41
Linalool	1086	1.28
Dimethyl tetrasulfide	1110	11.84
Trans-propenyl propyl disulfide	1117	2.38
Dipropyl trisulfide	1129	8.11
Methyl 2-propenyl trisulfide	1140	1.85
1,3,5-Trithiane	1156	0.35
Borneol	1168	2.89
Iso-Verbanol	1181	1.49
Safranal	1189	0.38
Tetradecanoic acid	1191	2.18
n-Decanal	1196	0.73
Neral (Z-Citral)	1229	6.20
Hexadecanoic acid	1257	7.77
Geranyl acetate	1371	1.52
Pentadecanoic acid	1380	2.61
β-Cubenene	1384	0.18
Pentacosane	1400	5.81
β-Caryophyllene	1433	1.23
β-Ionone	1477	0.76
γ-Cadinene	1515	0.29
Caryophyllene oxide	1572	4.33
Epi-α-Muurolol	1636	1.91
α-Cadinol	1659	2.06
n-Heptadecane	1684	0.13
Gereninal (E-Citral)	1765	3.60
(Z,Z)-9,12-Octadecadienoic acid	1819	1.08
Hexacosane	1848	0.93
Nonacosane	1906	0.62
Total identified		94.62

* RI, retention index. ** Normalized peak area abundances without correction factors.

**Table 2 molecules-27-07032-t002:** Effect of free and nanoliposomal A_EO_ color properties of Salep mucilage bionanocomposite films.

Samples	L	a	b	ΔE	WI	YI	Transparency (%)
Control film	57.33 ± 3.06 ^a^	0.51 ± 0.26 ^c^	−0.87 ± 0.42 ^c^	40.48 ± 3.06 ^c^	57.32 ± 3.06 ^a^	−2.16 ± 1.03 ^c^	62.71 ± 1.05 ^a^
A_EO_-loaded Film	41.00 ± 3.61 ^c^	10.90 ± 0.46 ^a^	20.30 ± 1.05 ^a^	61.31 ± 3.32 ^a^	36.64 ± 3.32 ^c^	71.01 ± 5.75 ^a^	40.32 ± 0.84 ^c^
NL_P_/A_EO_-loaded Film	49.67 ± 4.50 ^b^	3.13 ± 0.61 ^b^	6.13 ± 0.77 ^b^	48.63 ± 4.53 ^b^	49.19 ± 4.53 ^b^	17.87 ± 3.83 ^b^	56.01 ± 0.67 ^b^

For each column, means with superscripts (a–c) are significantly different (*p* < 0.05). Data are means ± SD.

**Table 3 molecules-27-07032-t003:** Effect of free and nanoliposomal A_EO_ on physical and mechanical properties of Salep mucilage bionanocomposite films.

Samples	Thickness (µm)	Contact Angle (°)	Moisture Content (%)	Water Solubility (%)	Tensile Strength (MPa)	Elongation at Break (%)
Control film	86.90 ± 2.19 ^b^	68.39 ± 3.42 ^c^	12.89 ± 0.12 ^a^	50.71 ± 1.48 ^a^	18.90 ± 1.23 ^a^	82.14 ± 2.88 ^b^
A_EO_-loaded Film	92.09 ± 1.33 ^c^	102.48 ± 3.71 ^b^	11.08 ± 0.17 ^b^	45.18 ± 1.22 ^b^	11.46 ± 1.15 ^c^	103.51 ± 3.60 ^a^
NL_P_/A_EO_-loaded Film	105.42 ± 2.51 ^a^	124.02 ± 3.37 ^a^	10.37 ± 0.08 ^c^	40.91 ± 1.62 ^c^	16.37 ± 1.02 ^b^	63.09 ± 2.74 ^c^

For each column, means with superscripts (a–c) are significantly different (*p* < 0.05). Data are means ± SD.

**Table 4 molecules-27-07032-t004:** Antibacterial activity (growth inhibition zone based on mm) of NL_P_, A_EO,_ and NL_P_/A_EO_.

Microbial Strain		Inhibition Diameter (mm)			
	Day	NL_P_ (Negative Control)	A_EO_	NL_P_/A_EO_	Chloramphenicol (Positive Control)
Staphylococcus aureus	3rd	6.7 ± 0.3	16.3 ± 0.5	14.5 ± 0.9	20.3 ± 0.5
	30th	6.3 ± 0.3	9.7 ± 0.7	12.4 ± 0.8	14.6 ± 0.8
Escherichia coli	3rd	6.5 ± 0.4	12.9 ± 0.9	10.3 ± 0.6	15.1 ± 1.1
	30th	6.2 ± 0.2	7.3 ± 0.6	9.8 ± 0.5	11.2 ± 0.9

## Data Availability

Data will be provided upon request.

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
