# Peer review of "Characterization and Optimization of Salep Mucilage Bionanocomposite Films Containing Allium jesdianum Boiss. Nanoliposomes for Antibacterial Food Packaging Utilization"

_molecules, 2022, doi:10.3390/molecules27207032_

Round 1

Reviewer 1 Report

The author has presented significant research displaying the antibacterial food packaging utilization. However, the author is suggested to check the manuscript thoroughly for formatting errors. Consider these points:

Kindly check the font size in line number 50-51, 106, 

Rewrite the sentence in line 102

Improve the conclusion section and emphasize more on its futuristic applications.

Reviewer 2 Report

The authors has described the development and characterization of nanoliposome-loaded Salep mucilage-based bio-nanocomposite films containing free and encapsulated Allium Jesdianum Boiss. essential oil (AEO) and evaluated their physical, mechanical and antibacterial properties. The results of current study are well described and support the work done. Few minor suggestions are recommended prior to be accepted for its publication.

1. In the introduction part, it is better to cover some relevant literature on nanoliposomes loaded essential oils for food packaging application.

2. At page 4, in section "Preparation of bio-nanocomposite films and characterization" the sentence "After that, the film-forming solution was poured into onto plastic Petri dishes" need to be corrected to "After that, the film-forming solution was poured onto plastic Petri dishes".

3. Equation 6, should also be mentioned in the text.

4. At page 7, in the section "Release Properties" the sentence "was determined by the following Eq. (2):" need to be corrected to "was determined by the following Eq. (9):".

Reviewer 3 Report

A general remark - the authors have put a lot of effort in developing the study and the manuscript. Unfortunately, the written document lack of coherence. Some specific observations:

- Use of abbreviations before their definitions (see for instance NLp, Line 16);

- Measurements units are not always indicated;

- Please reword Lines 13-15 for easier understanding;

- Line 25 - Please check if this is correct : "antibacterial stability"?

- Section 2.1. Materials - A lot of used materials are not included;

- Line 122: "lecithin: cholesterol, and AEO (3:1)" - the ratios are confusing; The ratio of each mixture component should be indicated;

Line 137: "Centrifugation methods were used to assess efficiency [17]." Please develop the content. Centrifugation is a separation operation. What do authors mean by "Centrifugation methods"?

Lines 164 - 166, again the ratios/concentrations are not clearly indicated. Please reword.

Line 242: Please explain "1489.8 Pa".

Line 250 - unusual measurement unit for O2P - Why use, in the same ecuation cm, microm and m? Use just one of them, or SI units;

Line 252 - explain "101 kPa";

Line 265 - indicate why 275 nm;

2.4.6. Add some information on positive control;

Fig. 5: Please explain A and B. "Effects of different ratios" is not clear. Please reword.

Table 4: Indicate measurements units for growth inhibitions zone.

Round 2

Reviewer 3 Report

The authors put a lot of effort in developing and improving the manuscript.